# Virtual Experience of Perioperative Patients: Walking in the Patients’ Shoes Using Virtual Reality and Blended Learning

**DOI:** 10.3390/ijerph18126457

**Published:** 2021-06-15

**Authors:** Hyeon-Young Kim, Ji-Hye Lee, Eun-Hye Lee

**Affiliations:** 1Nursing Department, College of Nursing, Sahmyook University, Seoul 01795, Korea; hyykimm@syu.ac.kr; 2VR Healthcare Content Lab, Sahmyook University, Seoul 01795, Korea; 3Nursing Department, Sahmyook Medical Center, Seoul 02500, Korea; ezlove@hanmail.net

**Keywords:** perioperative nursing, virtual reality, learning, nursing student, qualitative research

## Abstract

PURPOSE: This study examined the significance, nature, and structure of the virtual experience of perioperative patients as undergone by nursing students during their practical training through VR and blended learning. METHODS: Data were collected through a focus group interview (FGI) of 21 nursing student participants from November 2019 to December 2019 and analyzed through Colaizzi’s phenomenological method. RESULTS: Seven theme clusters were organized that described nursing students’ experiences. They are “placed in a passive position,” “facing the limits of communication,” “thinking of developing and improving competency as a nurse,” “recognizing the importance of interacting with their patients”, “learning vividly through experience”, “engaging in a new type of participatory learning”, and “designing nursing knowledge.” CONCLUSION: Patient-centered care can be achieved in the nursing school curriculum through “patient experiences.” Additionally, the feedback from research participants who have “become keenly aware of the need for patient experiences” shows that empathizing with the “patient experience” is an essential quality to acquire by prospective medical professionals before they are introduced to the nursing field. We suggest future studies that expand on nursing students’ patient experience in various teaching methods and curriculums.

## 1. Introduction

The Fourth Industrial Revolution induced changes in the use of innovative information and technology in the medical field. These changes, such as the emphasis on social responsibility in contributing to national health and the globalization of nursing, have moved the medical paradigm toward “patient-centeredness.” [1]. Patient-centeredness establishes partnerships between healthcare providers, patients, and patients’ families, with the goal of respecting the desires, needs, and preferences of patients in the decision-making process while ensuring that the necessary education and support are provided to enable patients to make decisions and participate in their own medical care [2]. Moreover, it is based on the recognition that the medical system should counter problems such as the invasion of dignity and autonomy, as well as the anxiety and humiliation patients may experience during treatment [3]. For this reason, the Health Insurance Review and Assessment Service evaluates patient experiences based on medical services or hospital environments on 2017 [4]. Furthermore, the patient experience refers to everything that a patient thinks, sees, hears, and feels in the process of being provided medical care, as well as prejudice or bias toward medical institutions and all the direct or indirect perceptions that occur during hospitalization and/or after discharge [5,6]. Additionally, this holistic concept that encapsulates medical and non-medical factors with psychological and cultural interactions has recently become the standard for qualitatively evaluating medical institutions [7]. Nurses in particular encounter high social demands and requests, as they are the medical professionals who provide medical care to patients at the closest proximity. It is therefore essential to understand clinical situations, and for the patient to adapt to the changing clinical medical environment, in providing high-level nursing care [8]. However, the direct experience of nursing for undergraduate nursing students, who are prospective medical professionals, is continuing to decline; in addition to the lack of hospitals that provide clinical education [9], the major courses do not reflect the shifting medical environment and field.

Virtual reality (VR) is an important technology emerging from the Fourth Industrial Revolution; it refers to a technology that allows users to feel and act as if they were in a specific situation by simulating that situation through a headset or physical equipment [10]. VR provides the advantage of complementing the limitations of existing mannequins or standardized patient simulations, and there have been increasing attempts to apply this method in the educational environment as an alternative or supplemental educational methodology [11,12]. Moreover, blended learning, also referred to as hybrid learning, is an educational method that enhances the learning effect by combining the advantages of the existing methods [13]. Blended learning reaches beyond incorporating online and offline learning methods and can extend the combination of various learning elements into a design strategy [14]. While there has been recent research on VR simulations and blended learning in general [15], application to the nursing field remains insufficient. 

Recently, a number of specialized nursing fields such as operating rooms, neonatal rooms, intensive care units, and delivery rooms have been closed to students due to patient safety, infection concerns, and a lack of cases [16]. Particularly, operating rooms have been undergoing strict infection control due to the decrease in patients’ immunity and the possibility of opportunistic infections [17], which has limited clinical placements for nursing students. However, because perioperative patients receive various medical treatments in the ward from hospitalization to discharge, in addition to their experience in operating rooms, an organic connection and an integrated understanding are necessary between surgical nursing and ward nursing to understand perioperative patients [18].

A number of studies have reported the theory and real-world effects of simulation education to date [19,20] however, studies from the perspective of patients who are not medical professionals are scarce. Currently, the operating room has become more inaccessible due to the COVID-19 pandemic, and the limitations of clinical practice are increasing.

VR can embody the operating room scene, which is difficult for nursing students to access, and allow nursing students to experience being a surgical patient through sights, hearing, and touch [11,12]. The present study analyzes the nursing student’s learning experiences and outcomes in a virtual experience of perioperative patient program, simulating a perioperative patient’s perspective for patient-centered care. This study examined the significance, nature, and structure of the virtual experience of perioperative patients as undergone by nursing students during their practical training through VR and blended learning. The study also aimed to understand nursing from the perspective of perioperative patients, and enhance the understanding of the integrative experiences that perioperative patients encounter. Moreover, this study intends to provide basic data to identify methods to improve nursing curricula.

## 2. Methods

### 2.1. Research Design

This is a phenomenological study designed to understand the meaning and nature of nursing students’ virtual experiences. Using VR and blended learning during practical education, the nurses engaged in a perioperative patient simulation.

#### 2.1.1. Research Participants and Researcher Preparation

Purposive sampling was used to collect nursing students’ virtual experiences of being a perioperative patient using VR and blended learning [21]. Among the second-year nursing students enrolled in one university located in city S, candidates who understood the purpose and intention of the study through a social network service (SNS) announcement voluntarily agreed to participate in the “patient experience” and focus group interviews. The recruitment announcement included the research topic, the research purpose, the participation procedure and method, the participation period, and the benefits. There were a total of 21 participants (13 female, average age 21.4 years). For the focus group interviews, the study was conducted by forming three groups consisting of seven participants each.

#### 2.1.2. VR Blended Learning Program

The program was conducted for five weeks, and consisted of the following four sessions: (a) Educational lectures, (b) Problem-based learning I (individual activities), (c) Problem-based learning II (team activities), and (d) Simulation using VR and wearable device. In accordance with the research design, the following program contents were applied to the nursing students: (a) (Pre-operative stage) To receive an intravenous injection while wearing an arm vein model, and to move with an IV Pole connected to a fluid bag and use the bathroom, (b) (Moving to the operating room stage) To try moving while lying on a stretcher, and to transfer to the operating room path while wearing the VR equipment, (c) (Waiting room stage) To expose and confirm the pre-marked breast surgery site while wearing a wearable breast model, (d) (Intraoperative stage) To change the patient’s position using the actual operating room bed, positioner, and restraints, and to experience the operating room environment while wearing the VR equipment, (e) (Postoperative stage) To undergo a simple catheterization procedure wearing a catheterization model.

#### 2.1.3. Research Procedure and Data Collection

Data collection was conducted from 28 November 2019 until 31 December 2019 of the year when the data became saturated. Focus group interviews were conducted twice per group at the participants’ preferred locations. Each interview was approximately 80 min to 120 min in duration. The interviews were conducted in easily accessible conference rooms or rest areas that were quiet and allowed participants to feel comfortable and free to talk. All interviews were audio recorded with the participants’ prior consent, and field notes were recorded by summarizing the interview content and the participants’ appearance, reaction, and atmosphere as observed during the interviews. For data analysis, a research assistant transcribed the recorded content exactly as it was recorded, and the researcher confirmed that it was entirely correct by comparing the audio recordings with the transcript.

Interview questions were open and semi-structured, with introductory, conversion, main, and closing questions. After starting the introductory questions with daily greetings and questions regarding the indirect experiences of being a patient, the interviewer naturally approached the research question using the following transition question: “How did you feel when you became a patient (how did you feel overall)?” After asking nine main questions including, “How was the virtual experience of being a perioperative patient through VR?” the interview concluded by asking for additional opinions.

During the interview, the participants were given the time and opportunity to sufficiently present their experiences and opinions, and the interview was conducted until there were no further comments. Ambiguous statements from participants underwent additional clarification procedures. After the first focus group interview, participants were individually contacted up to three times to ask further questions and/or seek clarification of content during the analysis when needed. The opinions of participants who voluntarily provided additional insights after the end of focus group interviews were also included in the data collection. In the second focus group interview, a summary and analysis of the first focus group interview was provided, and the content of this research topic was further explored in-depth.

### 2.2. Data Analysis

Colaizzi’s method emphasizes matching appropriate data sources and data collection methods, and focuses on discovering the meaning of empirical phenomena that the participants commonly state, rather than individual attributes experienced by the participants. This study was analyzed following Colaizzi’s study in order to focus on deriving the experiential attributes to which nursing students commonly refer regarding the virtual experience of being a perioperative patient [22]. The coding and categorization process was conducted using the program QSR NVivo 12 (QSR International PTY Ltd, Melbourne, Australia).

The participant statement transcripts were read repeatedly to understand the emotions and the overall meaning. Meaningful phrases or sentences related to their experiences were extracted from the statements, and similar or overlapping sentences were compared, analyzed, and converted into representative statements. Based on this process, meanings were constructed through restatement in the researcher’s language, and the derived meanings were organized into themes, clusters, and categories. At this stage, the categories were compared with the original data to review whether the content maintained the context initially stated by the participants. Finally, the fundamental structure of the research phenomenon was comprehensively described by integrating the analyzed content, followed by the process of conceptualizing the derived categories.

To increase the validity of the study, the analyzed results were presented to the participants in the interview, and the results were readjusted and supplemented after receiving feedback.

### 2.3. Research Rigor

To ensure the rigor of the qualitative research, efforts were made to increase the truth value, applicability, consistency, and neutrality, according to the evaluation criteria suggested by Lincoln & Guba [23]. Additionally, the COREQ (Consolidated Criteria for Reporting Qualitative Research) reporting guide was used to maintain transparency, systematic analysis, and evaluation during the research procedure [24].

### 2.4. Ethical Considerations

The study was approved by the university’s bioethics review committee (approval number: 2-7001793-AB-N-012019076HR). Prior to data collection, the researcher explained the research purpose, method, and process, as well as the participants’ rights, and written consent was received from those who voluntarily agreed to participate. Participants received further information that the interview content was solely for research purposes, confidentiality was guaranteed, and the audio files would be discarded immediately after the transcript was written. Interview data for qualitative analysis were de-identified before being stored. Participants were provided with token gift vouchers and refreshments at each interview.

## 3. Results

### 3.1. Categories and Theme Clusters

A total of 187 meaningful statements were derived from the original data collected, and 21 themes were composed through repeated verification with relevance to the study’s topic. According to the similarity of meanings, data were organized into seven theme clusters and structured into four categories (Table 1; Figure 1).

#### 3.1.1. Category 1: Transference into the Position of a Patient Who Receives Nursing Care

The experience of becoming “a real (literally)” patient involved being placed in the patient’s position and feeling their emotions. This includes being “placed in a passive position” and “facing the limits of communication” in the position of a patient, which is contrary to the position of being a member of the medical staff.

##### Being Placed in a Passive Position

Participants stated that when they became patients, they assumed a “passive position” while undergoing procedures and operations. As a patient, they experienced the emotions of patients who become sensitive to even minor stimuli and complained of feelings of helplessness and anxiety. They felt pressured, as a patient who was undergoing procedures alone, surrounded by medical staff. Participants also experienced feelings of shame and embarrassment when they were publicly exposing their body parts from a passive position:

“When I was being transferred, lying on the stretcher, I felt like I could not even move, though I was not actually sick … I felt like I was perhaps on my way to death, on my way to the grave. It was scary. I felt helpless.”

“I was looking at the ceiling with my arms out, and I felt I could not do anything … There was nothing that could be done (during urinary catheterization) … Even when I am watching the animation (VR) I am (really) looking at the ceiling … (I felt) pressure(d) from the feeling of being surrounded.”

##### Facing the Limits of Communication

As a result of being placed in a passive position as patients, the nursing students felt subjugated and had less authority. They were resigned to restricted communication due to the worry that the medical staff would withdraw from the emotional burden if they actively complained about discomfort or expressed their opinions:

“So, for me, I could not say some things to the medical staff when I was the patient. I felt that it could make the atmosphere harsh … because I am speaking with that person one on one. It made me think that if I say anything along the lines of “This does not sound right”, then they may not be so nice to me.”

“When I was being transferred on the stretcher, I felt conscious of (my own) weight, and I felt that if I said that whenever they moved me (made me uncomfortable), then they may not like me, and (they might) intentionally not do (the) things they should be doing for me.”

#### 3.1.2. Category 2: Understanding of and Insights into Nursing Competencies

The nursing students’ experience of being a patient was identified to provide an understanding of and insights into nursing competencies. The category was comprised of “thinking about developing and improving one’s competency as a nurse” and “recognizing the importance of interacting with patients.”

##### Thinking about Developing and Improving One’s Competency as a Nurse

After experiencing what it means to be a patient, the participants stated that they reflected on developing and improving their competencies as nurses. They stated that they recognized the need to provide skilled nursing care based on evidence as a professional and that they should pursue nursing with that in mind. Additionally, nursing students who focused on practicing and evaluating nursing skills experienced a visual confrontation involving being in the position of the patient, who is the recipient of nursing skills, and reflected on the nurse’s attitude, which should be combined with nursing skills. The participants demonstrated their determination to be better nurses through such reflection:

“During procedures, I was always busy, but being in the patient’s position, I felt quite strongly that even just saying a (kind) word could make them much less anxious.”

“It (was an opportunity for me to realize that perioperative patients are family members. The experience of being a patient encouraged me to think quite frequently about the term “responsibility.”

“If I consider the patients as my family, would I make a mistake? Wouldn’t I double check? That is what I was thinking. I want to provide better care.”

##### Recognizing the Importance of Interacting with Patients

The experience of being a patient helped the participants understand the patient’s position, their language, and the behavior that the patient wants. Moreover, the participants revealed that they were able to see patients as recipients of nursing care and as human beings. The participants recognized the need for a nursing practice that includes mutual interactions with the care recipient, and they reflected on the interactions between patients, nurses, and other medical staff:

“Honestly, I felt very different from when I (was the one doing) the procedures. I thought that I could be a politer nurse, if I remember that the patient is also a human being, like me.”

“I guess it is like respecting (the patient) as a human being. I never thought about (being) compassionate, but through this experience, I realized that patients need compassion, and it helped me consider them, once again, as human beings.”

“I thought, “Maybe patients need to know that they have right to ask questions and things like that.” That is because they are all patients for the first time, they are undergoing procedures for the first time, so they would not be able to speak properly due to fear and anxiety.”

#### 3.1.3. Category 3: Enhancement of Academic Achievement through New and Vivid Teaching Methods

In this category, new and vivid teaching methods such as VR and blended learning were introduced during the process of becoming a patient in practical nursing education. Participants experienced enhanced academic achievements that included two themes: “learning vividly through experience” and “engaging in a new type of participatory learning.”

##### Learning Vividly through Experience

The blended education method of learning through VR provided the study participants with vivid practical training, allowing them to learn the overall surgical process from an abstract perspective connected to reality. Through an unexpected surgery scene and the preparation process, participants had new and powerful encounters in the unpredictable nursing field and were motivated and inspired to receive practical education in nursing:

“Oh, the operation theater (in VR) was darker than I thought. I thought it would be brighter.”

“(The experience) made me realize that patients feel very uncomfortable in (ways) I had never thought about. I had to go to the toilet with an IV pole … and getting dressed was also uncomfortable … I felt a lot of (unexpected) discomfort (while doing) things I had never (thought much about before).”

“The hips should be pulled forward as far as possible. That posture was very difficult. It was so embarrassing when I had to wear the equipment. And when they inserted the IV, it really hurt.”

##### Engaging in a New Type of Participatory Learning

The participants were able to observe and experience the surgical process from the first-person perspective of a patient, using VR. Using the wearable VR model, participants had the virtual experience of a perioperative patient, combining the body’s sensory reactions with personal thoughts:

“People were crowded on both side of the chairs (in VR)… There were many people on my way to the operation theater, and I felt I would become more anxious … That was (the) most memorable. Dramas and movies only depict scenes during surgery; (they don’t) quite show what the theater looks like, but I was able to see 360 degrees around, and there were things like a kidney basin and gauze with blood on it. I saw timers with anesthesia time and operation time. I also saw medical devices. So, it was interesting.”

“I thought there would probably be things like the surgical lights or Bovies, which are the basic items, but I never knew that they check the time. (I learned that during) the VR, and it felt like the timer was quite an important device, so that was interesting.”

#### 3.1.4. Category 4: Application of Patient-Centered Care

During the virtual experience of being a perioperative patient, deep contemplation about the application of nursing for patient-centered care was categorized into a theme cluster called “designing nursing knowledge” toward developing new methods to deliver nursing care.

##### Designing Nursing Knowledge

Through this experience, the participants realized for the first time that nursing knowledge needs to be appropriately transformed or reorganized according to the nursing situation or the care recipient’s condition. Therefore, the participants underwent “designing nursing knowledge.” The participants became keenly aware that the patient’s experience must be included in the nursing education curriculum. Additionally, they suggested sensitivity to gender-perspectives by considering the patient’s position when they receive nursing care. Through practical education using VR and blended learning, participants produced creative and effective nursing service ideas and devised plans for patient-centered nursing care:

“I came to think that (understanding the emotions involved) in the patient’s experience is very important for medical staff. Things like posture were very difficult to rectify. I think it would be easier for the patient and the staff if there is (a) marking to show where to enter. Maybe something like, “Please lower your backside until the red point.”

“(Since) the patients I care for can be (either) males or females, it was an opportunity for me to make a sensitive consideration, as I gained experience and (thought) about those things in advance.”

“I felt worse than I expected when I was being moved onto the bed. I had severe nausea when I was being moved and when they were using the bed sheet (because) the staff could not control their strength very well.”

“So, I thought that during urinary catheterization, (rather than undressing), it would be better to cover everything (and) just use the insertion point … so that’s the only part that’s visible to cover and let it be seen or cover it to help the patient feel better.”

“I think there must be medical staff to (attend to) the (patient’s) psychological aspects.”

## 4. Discussion

This study used a phenomenological method by conducting focus group interviews with nursing students who participated in practical education using VR and blended learning. The purpose was for undergraduate nursing students to understand the meaning and essence of the patient experience during practical education. Nursing students have limitations in direct patient care during practical education; therefore, undertaking the patient’s role as a patient during practical education is a method of indirect experience of patient care and a useful experience for a prospective nurse [25].

A major contribution of this study was in enabling nursing students to recognize the patient’s position as an important measure for patient-centered medical care [26]. Although via an indirect “patient experience”, the VR environment allowed nursing students to experience the perioperative patient’s perspective. Therefore, this curriculum can effectively help nursing students understand the patient’s position. Previous studies on negative experiences with nurses with regard to discharged patients are consistent with the “patient experience” of nursing students in this study, including avoidance, neglect, lack of explanation, and discrimination [3]. Facing the “limit of communication” from the patient’s view point is another important factor for the patient-centered medical field. Based on this, the opportunity for “patient experience” to consider and understand the patient’s position should be included in undergraduate nursing students’ curriculum, and the need to provide patient-centered care should be emphasized.

Through the “patient experience”, the “understanding and insight into nursing competencies” factor allowed participants to contemplate evidence-based nursing and the nurse’s attitude as important factors in becoming better nurses. Such realizations are considered to be starting points in the formation of a positive professional nursing intuition. Within undergraduate nursing student education, in which the formation of the correct professional nursing intuition begins [27], the “patient experience” as part of patient-centered medical care will be indispensable in improving education quality.

Reflecting on interactions from the patient’s perspective and coming to understand patients as the human recipients of care will improve the quality of care in the dimension of the nursing metaparadigm. Since participants were able to gain a practical understanding of nursing through practical learning, the “patient experience” should be emphasized in order to pursue the essential meaning of nursing studies that supports human well-being.

Education using VR has gradually been on the rise in recent years, and the fact that it is effective in encouraging empathy and intimacy, along with its convenience [28,29], is in line with the results of this study’s “vivid teaching method.” In terms of vivid pedagogy, “new and powerful encounters with the nursing field” will be a useful approach to practical education for nursing students [30]. The virtual experience of being a perioperative patient through VR and blended learning is a participatory form of class education that allows students to understand the nursing phenomena through their bodies and actually experience nursing. Therefore, it will help students understand the field and lay the foundation to strengthen core competencies. Moreover, blended learning, including VR, should be actively considered as a pedagogy in practical education to overcome the limitations of field practice, which has restrictions with respect to direct patient contact.

Furthermore, “designing nursing knowledge” will act as an effective competency for individual nurses to implement key factors that will deliver coping abilities and efficient nursing care in the ever-changing nursing field by empathizing with the patient’s position. Based on these educational effects, there is a need to carefully pursue the inclusion of patient experience in nursing students’ practical education curriculum [31].

This study investigated the meaning and essential structure of nursing students’ experience with patients through focus group meetings after practical training using VR and blended learning. Therefore, it presents basic data to incorporate the “patient experience” category into nursing students’ regular curriculum, including their practical education. Additionally, because this study revealed and analyzed the meaning of being a perioperative patient for nursing students through a virtual experience, it can be differentiated from existing studies on patient experience that have mainly addressed clinical fields and diseases. Based on the results, future studies should expand the patient experience for nursing students into various pedagogies, subjects, and curricula. Moreover, observation and physical experience ‘in the body’ from a first-person perspective through VR and blended learning can be interpreted as a “physical phenomenological bodily experience”, which is a perceptual experience through the cognitive thinking process. Furthermore, this study applied Colaizzi’s [1978] [22] phenomenological analysis method to understand the essential structure of the experience of being a patient. Based on the results of this study, which demonstrate that the body’s sensory responses and cognitive thinking go hand in hand, the use of Merleau-Ponty’s “phenomenology of body” is also proposed [32].

The limitation of this study is the restricted sample size, with participants only recruited from one university. It is important to expand the study results in the future by including a variety of subjects.

## 5. Conclusions

This was a phenomenological study that was conducted to confirm the meaning, nature, and structure of nursing students’ virtual experience of being a perioperative patient using VR and blended learning during practical education. The categories identified through this study regarding the virtual experience of being a perioperative patient included “transference of the patient’s position”, “gaining an understanding of and insights into nursing competencies”, “the enhancement of academic achievement through new and vivid teaching methods”, and “the application of patient-centered care.” Evidently, being in the patient’s shoes, as it were, inspired an understanding of patients as the human recipients of care, which is one of the four nursing metaparadigms. The results of this study also identified that the perception of patient-centered medical care can be achieved through the patient experience within the nursing curriculum. Additionally, through feedback from the research participants, who “became keenly aware of the need for patient experience”, it was revealed that patient experience is an essential element that must be acquired by any prospective medical practitioner before undertaking direct experience in the nursing field.

Given that the emphasis on the importance of patient-centered medical care has been growing, nursing colleges should seek to develop competencies in undergraduate nursing students by including patient experience in the nursing curriculum. This study has shown that there is a need for continuous research on the development of nursing competencies in undergraduate nursing students through patient experiences, as well as studies on methods for effectively including the patient experience in the curriculum. Research is also required to evaluate nursing students’ competency development for patient-centered care through the experience of being a patient.

## Figures and Tables

**Figure 1 ijerph-18-06457-f001:**
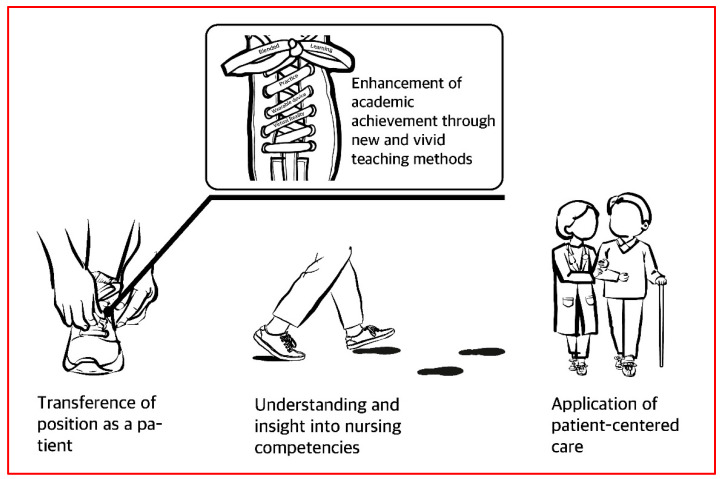
Walking in the Patients’ Shoes: Virtual Experience of Perioperative Patients.

**Table 1 ijerph-18-06457-t001:** Virtual experience of perioperative patients.

Category	Theme Cluster	Theme
Transference of position as a patient	Placed in a passive position	Complains of helplessness and anxiety
		Becomes sensitive to even minor stimuli
		Feels shame and embarrassment
		Experiences pressure
	Facing the limits of communication	Wants the minimum amount of consideration as the patient
		Experiences a feeling of dejection due to one-way and limited communication
Understanding and insight into nursing competencies	Thinking of developing and improving competency as a nurse	Comes to understand and pursue evidence-based nursing as a nurse
		Thinks about nurses’ attitudes
		Is determined to become a better nurse
	Recognizing the importance of interacting with their patients	Experiences what it is like to be a patient
		Comes to view the patients as human beings who need care
		Thinks seriously about the interaction between the patient and medical staff
Enhancement of academic achievement through new and vivid teaching methods	Learning vividly through experience	Learns the surgery process realistically rather than abstractly
		Faces unexpected an nursing field in a new and powerful way
		Learning is motivated, and curiosity is spiked
	Engaging in a new type of participatory learning	Experiences VR
		Experiences VR through a wearable device
Application of patient-centered care	Designing nursing knowledge	Is keenly aware of the need for patient experience
		Thinks about gender-perspective sensitivity
		Comes up with creative and effective nursing service ideas
		Devises a plan for patient-centered care

## Data Availability

The data presented in this study are available on request from the corresponding author.

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
