# Peer review of "Virtual Experience of Perioperative Patients: Walking in the Patients’ Shoes Using Virtual Reality and Blended Learning"

_ijerph, 2021, doi:10.3390/ijerph18126457_

Round 1
Reviewer 1 Report
In this paper, the authors conducted a study on the Virtual Experience of Perioperative Patients. I have the following concerns.
- What are the role of blending learning and VR in this study, respectively. The authors should clarify this.
- What is the purpose and motivation of this study?
Author Response
We would like to express our appreciation for your extremely thoughtful suggestions. Your feedback was extremely helpful to strengthen our manuscript. As you will see below, we have been able to revise and improve the paper as a result of your valuable feedback.
Overall, we have made changes throughout the paper that address the points you have made as shown below. The corrected parts can be check in red and indicated with page numbers in the table below for easy reference.
Thank you again for taking the time to share your constructive feedback.
Yours sincerely,
The authors
|
Reviewer Comment |
Author Response to Comment |
Changes made to Article |
Page |
|
Reviewer: 1 |
|
|
|
|
Point 1
1. What are the role of blending learning and VR in this study,respectively. The authors should clarify this. |
As per your comment, we added the research method regarding the VR and blended learning in the Methods sections. |
None è 2.1.2. VR blended learning program The program was conducted for five weeks, and consisted of the following four ses-sions: (a) Educational lectures, (b) Problem-based learning I (individual activities), (c) Problem-based learning II (team activities), and (d) Simulation using VR and wearable device. In accordance with the research design, the following program contents were ap-plied to the nursing students: (a) (Pre-operative stage) To get an intravenous injection while wearing an arm vein model, and to move with an IV Pole connected to a fluid bag and use the bathroom, (b) (Moving to the operating room stage) To try moving while lying on a stretcher, and to transfer to the operating room path while wearing the VR equipment, (c) (Waiting room stage) To expose and confirm the pre-marked breast surgery site while wearing a wearable breast model, (d) (Intraoperative stage) To change the patient’s posi-tion using the actual operating room bed, positioner, and restraints, and to experience the operating room environment while wearing the VR equipment, (e) (Postoperative stage) To undergo a simple catheterization procedure wearing a catheterization model. |
3
|
|
Point 2
2. What is the purpose and motivation of this study? |
As per your comment, we amended and added the Introduction. Virtual experience is derived from the teaching method of deliver the perioperative patient experience more effectively. We remove the phrase to make the reader confused (delete the “to fill the research gap”). For the clarification the purpose of the article, we added the sentence in the introduction session regarding of the program method and modify the sentence as you mentioned. |
To fill this research gap, the present study analyzes the learning experiences and outcomes of a virtual experience program simulating a perioperative patient’s perspective. The effects of blended learning, including VR, are analyzed qualitatively with focus group interviews with the goal~ è VR can be embodied the operating room scene, which is difficult for nursing students to access, and allowed nursing students to experience being a surgical patient through sights, hearing, and touch. [11, 12]. The present study analyzes the nursing student’s learning experiences and outcomes of a virtual experience of perioperative patient program simulating a perioperative patient’s perspective for patient-centered care. This study examined the significance, nature, and structure of the virtual experience of perioperative patients as undergone by nursing students during their practical training through VR and blended learning. Also, the study aimed to understand ~ |
2 |

Reviewer 2 Report
This study evaluates the use of virtual experience of perioperative patients for nursing students as part of their practical training.
Line 34: Change "countermeasure" to "counter."
In the last two paragraphs of the Introduction you point out that the gap in the literature filled by this study is a lack of perioperative patients' perspective of simulated learning but under 2.1. Research design you state that you are looking at nursing students virtual experiences. These appear to be conflicting so I suggest rewriting these few paragraphs to make them clearer. i think the first sentence in line 76 should state clearly that it was the nursing students that were experiencing the VR as patients to get a patients perspective.
Lines 100-101: You mention that the data was collected until 31st December in the year of data saturation. Please include the year.
The Discussion is well written and clearly explains the importance of this research and how VR technology of the patent experience for nursing students can be implemented to enhance patient-centred care.
Author Response
We would like to express our appreciation for your extremely thoughtful suggestions. Your feedback was extremely helpful to strengthen our manuscript. As you will see below, we have been able to revise and improve the paper as a result of your valuable feedback.
Overall, we have made changes throughout the paper that address the points you have made as shown below. The corrected parts can be check in red and indicated with page numbers in the table below for easy reference.
Thank you again for taking the time to share your constructive feedback.
Yours sincerely,
The authors
|
Reviewer Comment |
Author Response to Comment |
Changes made to Article |
Page |
|
Reviewer: 1 |
|
|
|
|
Point 1
1. Line 34: Change "countermeasure" to "counter." |
As per your comment, we’ve corrected the sentence. |
Moreover, it is based on the recognition that the medical system should countermeasure problems è Moreover, it is based on the recognition that the medical system should counter problems |
1
|
|
Point 2
2. In the last two paragraphs of the Introduction you point out that the gap in the literature filled by this study is a lack of perioperative patients' perspective of simulated learning but under 2.1. Research design you state that you are looking at nursing students virtual experiences.
These appear to be conflicting so I suggest rewriting these few paragraphs to make them clearer. i think the first sentence in line 76 should state clearly that it was the nursing students that were experiencing the VR as patients to get a patients perspective. |
As per your comment, we amended and added the Introduction. Virtual experience is derived from the teaching method of deliver the perioperative patient experience more effectively. We remove the phrase to make the reader confused (delete the “to fill the research gap”). For the clarification the purpose of the article, we added the sentence in the introduction session regarding of the program method and modify the sentence as you mentioned. Also, According to your comment, we added the subject who were experienced to get the perioperative patient’s perspective. |
To fill this research gap, the present study analyzes the learning experiences and outcomes of a virtual experience program simulating a perioperative patient’s perspective. The effects of blended learning, including VR, are analyzed qualitatively with focus group interviews with the goal~ è VR can be embodied the operating room scene, which is difficult for nursing students to access, and allowed nursing students to experience being a surgical patient through sights, hearing, and touch. [11, 12]. The present study analyzes the nursing student’s learning experiences and outcomes of a virtual experience of perioperative patient program simulating a perioperative patient’s perspective for patient-centered care. This study examined the significance, nature, and structure of the virtual experience of perioperative patients as undergone by nursing students during their practical training through VR and blended learning. Also, the study aimed to understand ~ |
2 |
|
Point 3
3. Lines 100-101: You mention that the data was collected until 31st December 2019 in the year of data saturation. Please include the year |
According to your recommendation, we corrected the sentence. |
Data collection was conducted from November 28th, 2019 until December 31st, è Data collection was conducted from November 28th, 2019 until December 31st, 2019 of the year |
3 |

Reviewer 3 Report
Thank you for the opportunity to review the original article Virtual Experience of Perioperative Patients: Walking in the Patients’ Shoes Using Virtual Reality and Blended Learning.
I appreciate the interesting and important topic of this study.
The article has potential but needs improvement in some sections.
Comments and suggestions for the Authors:
Abstract
The abstract contains basic information, keywords are corresponding with the content of the manuscript.
The aim of the study should be clearly stated (the same as in the main text).
Introduction
I think it would be important to mention in this section, why did you choose perioperative patients? and why this group of patients is worthy of attention?
The Authors should more clearly define the purpose of the work at the end of this section.
Material and methods
The methods selection and execution are consistent with the type of empirical study.
I only suggest adding a more detailed explanation of Colaizzi’s method.
Results
The results are clearly presented, the description is synthetic and confirmed by the presented data.
Discussion
The quality of the discussion sections is quite satisfactory, however adding some positions of recent literature (5-6 positions) could improve its overall level.
Some interesting studies to cite: PMID: 32930664, PMID: 31660522, PMID: 30271295
The Authors should also mention what the main limitations of the study are.
Conclusions
The conclusions are adequate to obtained results; especially practical implications for nursing colleges are important.
References
This section needs to be corrected according to the instruction for the authors.
Acknowledgments should be added at the end. Maybe would be worth acknowledging to participants of the study.
Author Response
We would like to express our appreciation for your extremely thoughtful suggestions. Your feedback was extremely helpful to strengthen our manuscript. As you will see below, we have been able to revise and improve the paper as a result of your valuable feedback.
Overall, we have made changes throughout the paper that address the points you have made as shown below. The corrected parts can be check in red and indicated with page numbers in the table below for easy reference.
Thank you again for taking the time to share your constructive feedback.
Yours sincerely,
The authors
|
Reviewer Comment |
Author Response to Comment |
Changes made to Article |
Page |
|
Reviewer: 3 |
|
|
|
|
Point 1
1. Abstract The abstract contains basic information, keywords are corresponding with the content of the manuscript. The aim of the study should be clearly stated (the same as in the main text). |
According to your comment, we’ve modified the purpose of this study in the manuscript introduction section to maintain the consensus with the abstract. |
To fill this research gap, the present study analyzes the learning experiences and outcomes of a virtual experience program simulating a perioperative patient’s perspective. The effects of blended learning, including VR, are analyzed qualitatively with focus group interviews with the goal~ è VR can be embodied the operating room scene, which is difficult for nursing students to access, and allowed nursing students to experience being a surgical patient through sights, hearing, and touch. [11, 12]. The present study analyzes the nursing student’s learning experiences and outcomes of a virtual experience of perioperative patient program simulating a perioperative patient’s perspective for patient-centered care. This study examined the significance, nature, and structure of the virtual experience of perioperative patients as undergone by nursing students during their practical training through VR and blended learning. Also, the study aimed to understand ~ |
2
|
|
Point 2
2. Introduction
I think it would be important to mention in this section, why did you choose perioperative patients? and why this group of patients is worthy of attention?The Authors should more clearly define the purpose of the work at the end of this section. |
As per your comment, we added the sentence to explain the current clinical training situation of nursing students regarding of the attention the perioperative patients. |
A number of studies have reported the theory and real-world effects of simulation education to date (17, 18) however, studies from the perspective of patients who are not medical professionals are scarce. è A number of studies have reported the theory and real-world effects of simulation education to date [19, 20] however, studies from the perspective of patients who are not medical professionals are scarce. Currently, the operating room has become more inaccessible due to the COVID-19 pandemic, and the limitations of clinical practice are increasing. |
2 |
|
Point 3
3. Material and methods The methods selection and execution are consistent with the type of empirical study. I only suggest adding a more detailed explanation of Colaizzi’s method. |
According to your recommendation, we’ve added the detailed explanation of Colaizzi’s method in the front of Data analysis section. |
None. 2.2. Data analysis è Colaizzi's method emphasizes matching appropriate data sources and data collection methods, and focuses on discovering the meaning of empirical phenomena that the participants commonly state rather than individual attributes experienced by the participants. |
4 |
|
Point 4. 4.Discussion The quality of the discussion sections is quite satisfactory, however adding some positions of recent literature (5-6 positions) could improve its overall level. Some interesting studies to cite: PMID: 32930664, PMID: 31660522, PMID: 30271295 |
Thanks for your exquisite reviewing for our article. According your recommendation, we’ve added the reference in the manuscript as ref no.12, 15, 19. |
è VR provides the advantage of complementing the limitations of existing mannequins or standardized patient simulations, and there have been increasing attempts to apply this method in the educational environment as an alternative or supplemental educational method [11, 12]. è While there has been recent research on VR simulations and blended learning in general [15], application to the nursing field remains insufficient. è Education using VR has gradually been on the rise in recent years, and the fact that it is effective in encouraging empathy and intimacy, along with the convenience of the edu-cation method [28, 29], |
2
2
9 |
|
Point 4. 4.Discussion The Authors should also mention what the main limitations of the study are. |
According to your comment, we’ve added the study limitation in the end of the discussion. |
è The limitation of this study is the restricted sample size, with participants only recruited from one university. It is important to expand the study results in the future by including a variety of subjects. |
10 |
|
Point 5. References This section needs to be corrected according to the instruction for the authors.
|
We’ve organized and corrected the reference list. |
è Please refer the reference in the manuscript. |
10~11 |
|
Point 6. Acknowledgments should be added at the end. Maybe would be worth acknowledging to participants of the study. |
According to your comment, we’ve added the study acknowledgement. |
è Acknowledgement: We sincerely appreciated the nursing students who voluntarily participated in the study and promptly shared their experiences |
10 |

Round 2
Reviewer 1 Report
The authors solved all my concerns. The paper can be published as the current form.